# Impact of the Nutrition–Inflammation Status on the Functionality of Patients with Chronic Kidney Disease

**DOI:** 10.3390/nu14224745

**Published:** 2022-11-10

**Authors:** Ángel Nogueira, Graciela Álvarez, Guillermina Barril

**Affiliations:** Department of Nephrology, Hospital Universitario de la Princesa, 28006 Madrid, Spain

**Keywords:** advanced chronic kidney disease, physical performance battery, nutrition, inflammation, malnutrition-inflammation score

## Abstract

Functional capacity of chronic kidney disease (CKD) patients is compromised by their nutrition-inflammation status. We evaluated the functional capacity of advanced chronic kidney disease (ACKD) patients and the influence of the nutrition-inflammation status. In a cross-sectional study, which included ACKD patients from the nephrology department of the Hospital Universitario de la Princesa in Madrid, Spain, we assessed: functional capacity with the Short Physical Performance Battery (SPPB) test, interpreting a result <7 in the test as low functionality; body composition with monofrequency bioimpedance; muscular strength with hand grip strength; nutritional and inflammatory status using biochemical parameters and the Malnutrition Inflammation Scale (MIS). A total of 255 patients with ACKD were evaluated, 65.8% were men, their mean age was 70.65 ± 11.97 years and 70.2% of the patients had an age >65 years. The mean score of SPPB was 8.50 ± 2.81 and 76.4% of the patients presented a score ≥7, with a higher percentage in the group of men. The percentage of patients with limitations increased with age. The patients with SPPB values higher than 7 showed high values of albumin and low soluble C-reactive protein (s-CRP) and MIS. We found better functionality in well-nourished patients. A multivariate logistic regression model established an association of high albumin values with a better functional capacity (OR: 0.245 CI: 0.084–0.714 *p* < 0.010), while another model showed an association between CRP values and decreased functionality (OR: 1.267 CI: 1.007–1.594 *p* = 0.044). Conclusion: nutritional status and body composition influence on the functional capacity of patients with ACKD.

## 1. Introduction

Currently, the prevalence of chronic kidney disease (CKD) has increased considerably, mainly in those subjects older than 65 years [1,2,3].

Therefore, in patients with CKD, the complications associated with this disease are greater comorbidity secondary to age, such as sarcopenia, which causes a decrease in functional capacity, as well as an increase in dependence, and secondary to this an increase in the risk of falls (and therefore an increased risk of bone fractures), and a higher number of admissions with a longer hospital stay, which altogether result in lower quality of life and an increased risk of mortality [3,4,5,6,7,8].

The decrease in functional capacity is caused by a decrease in muscle mass, which in the CKD patients may be caused by multiple factors such as: 1. increase in the concentration of uremic toxins (trimethylamine oxide, p-cresyl sulfate, indoxyl sulfate, phosphates, urea, β2 microglobulin), glycation end products that lead to metabolic and hormonal alterations (vitamin D and erythropoietin deficiency, insulin resistance, metabolic acidosis), which in turn may promote cognitive impairment, or malnutrition [9,10,11]; 2. decreased appetite leading to a deficit in caloric and protein intake that favors malnutrition [12,13]; 3. systemic inflammation resulting in elevated levels of inflammatory cytokines, such as interleukin 6 (IL-6), interleukin 1 beta (IL-1β), tumor necrosis factor (TNF-α), IL-1 receptor antagonist, and C-reactive protein (CRP) [9,10,11]; 4. comorbidities associated with CKD such as diabetes mellitus (DM), cardiovascular disease (CVD), or obesity [11]; 5. a sedentary lifestyle (greater muscle loss is observed in elderly and sedentary patients) [5,11].

Although these alterations can be observed at any stage of kidney disease, they are more frequent in predialysis stages, especially associated with protein energy wasting (PEW) [14].

The presence of PEW favors the decrease in functionality and is associated with muscle loss, in addition to other previously mentioned factors such as hormonal imbalances (including insulin resistance, alterations in TSH, decreased synthesis of erythropoietin, among others), systemic inflammation, increased catabolism, the release of myocytokine, or retention of toxins due to uremic syndrome [15].

In kidney disease, PEW causes loss of muscle mass, strength, and functionality (sarcopenia) in elderly patients, which are added to the changes caused by associated comorbidities; in addition, other compartments such as fat mass and cell mass may also be affected [16].

The objective of the study was to assess the functional capacity of patients with advanced chronic kidney disease (ACKD) and its relationship with their nutrition–inflammation status.

## 2. Materials and Methods

### 2.1. Study Design and Participants

A total of 225 patients with ACKD (Stages 3B, 4 and 5 (non-dialysis)) from a multidisciplinary unit of the nephrology service of the Hospital Universitario de la Princesa in Madrid Spain were evaluated.

This is an observational study, making a cross-sectional analysis of the patient database. The study was approved by the ethics committee of the Hospital Universitario de la Princesa, with registration number 4257.

Patient’s data were collected from their clinical history.

### 2.2. Evaluation of Functional Capacity and Muscle Strength

The functional capacity was determined by the Short Performance Physical Battery (SPPB) test, which assesses the functional capacity based on balance, gait speed, and lower limb resistance force, using 3 short subtests (Figure 1).

1. Balance test: it assesses whether the patient can maintain balance in 3 different positions: feet together, in semi-tandem, and in tandem, having to maintain each of the postures for a minimum of 10 s.

2. Four–meter test (it studies gait speed): it consists of measuring the time that the patient needs to walk 4 m. The test is performed two and even three times, collecting the data of the best time obtained.

3. STS5 test (it studies the strength and resistance of the lower limbs): the patient, with his/her arms, crossed over the chest and from a sitting position in the chair, must get up and sit down as fast as possible (the chair must have a height between 43–45 cm) [17].

Each of the subtests is scored from 0 to 4 depending on the time required to perform the test. The total score is the sum of the points obtained in each subtest. The functional capacity of the patients is categorized according to the SPPB score as follows:0–3 points, severe limitations3–6 points, moderate limitations7–9 points, slight limitations10–12 points, minimum/no limitations [17]

### 2.3. Frailty

Frailty was determined using the Fried criteria, which assess five items including unintentional weight loss, patient’s fatigue assessed through 2 questions from the Center for Epidemiological Studies Depression Scale (CES-D), muscle weakness assessed with dynamometry data adjusting the result for age and sex, decrease in gait speed, and decrease in the physical activity carried out by the patient.

To diagnose frailty, at least 3 of these criteria must be met [18].

### 2.4. Hand Grip Strength (HGS)/Dynamometry

Hand grip dynamometry determines the muscular strength of the upper limbs. To make the measurement, the dynamometer must be adjusted to the size of the hand, for a comfortable grip, performing it with the patient standing in an anatomical position and with the arm in which the measurement is made forming an angle of 90°. This measurement can be performed on both the dominant and non-dominant arm, making 3 measurements, leaving a time of about 30 s between measurements. The mean of the 3 values, the mean of the two highest measures, or the maximum value of the three measures can be used as a result; in this study, we considered the maximum value [19,20]. We used the hand dynamometer Baseline^®^ model 12-0240.

### 2.5. Assessment of Nutritional Status

Nutritional status was determined with the following parameters:Age, sex, and CKD-EPI as modifying factors of nutritional parameters.Nutritional study with classic laboratory nutritional parameters: albumin, prealbumin, CRP, lymphocytes, transferrin. Laboratory results were retrospectively collected from medical records. S-albumin was measured by the colorimetric standard method (Roche/Hitachi 904^®^/Modular ACN413) using the bromocresol green method s-Prealbumin and s-CRP were measured by immunoturbidimetry methods (Roche/Hitachi 904^®^/Model P: ACN 218, Roche Diagnostics, Basel, Switzerland) [21].Malnutrition Inflammation Score (MIS), is a scale that includes components of the subjective global assessment scale, combining them with body mass index, serum albumin, and transferrin. Each of the 10 components of the scale is graded according to severity (from normal to very severe), and the sum of the score of the 10 components classifies the patients according to their degree of malnutrition [22].

### 2.6. Body Composition Assessment

The body composition was determined by electrical bioimpedance (BIA), for this assessment a monofrequency bioimpedance apparatus was used (50 kHz and 800 μA, Body Impedance Analyzer BIA-101, Akern-RJL systems, Florence, Italy). Electrical bioimpedance is a method of estimating hydration, body composition parameters, and nutritional status. It is a safe, non-invasive, and non-observer-dependent technique. Bioimpedance measures the opposition exerted by the different cells and tissues of the organism to the passage of an alternating electric current. To perform this measurement, the patient is placed in a supine position with the arms separated by about 30° and the legs separated by about 45°. To close the circuit, two pairs of electrodes are placed, one on the back of the hand and the other on the back of the foot, separated by about 5 cm [23,24,25].

### 2.7. Statistical Analysis

Descriptive analysis was performed using absolute frequencies and percentages in the case of qualitative data, and by means ± standard deviation (SD) in the case of quantitative data.

A normality study of the quantitative data was carried out using the Kolmogorov–Smirnov test, although, due to the sample size, parametric tests were used, since they provide greater statistical power.

The comparison of quantitative data between groups was performed using Student’s *t*-test. Qualitative data were compared using the chi-square test or Fisher’s exact test.

Univariate and multivariate logistic regression models were performed to determine the nutrition–inflammation factors that influence the decrease in functionality. To do this, each factor was first compared with functional decrease, and those factors that were significant or close to significance (*p* < 20), were used in a logistic regression model by steps.

All statistical tests were considered bilateral, those results with *p*-values lower than 0.05 were considered significant. The data were analyzed with the statistical program SPSS version 23.0 (IBM Corp., Armonk, NY, USA).

## 3. Results

### 3.1. Characteristics of the Study Population

A total of 231 patients with ACKD were considered, although six patients could not perform the SPPB test and were excluded from the study (Figure 2).

Of the 225 patients included, 65.8% were men. Their mean age was 70.65 ± 11.97 years (median 73), and 70.2% of patients had an age >65 years.

Although the mean age of women was higher (72.19 ± 13.33 years) compared to that of men (69.85 ± 11.16 years) the difference was not statistically significant (*p* = ns).

Patients were classified according to age ranges, based on the classification of the latest consensus of the International Psychogeriatric Association (IPA), obtaining five age groups (<55 years, 55–64 years, 65–74 years, 75–84 years, and ≥85 years). The mean age of patients <65 years was 55.29 ± 7.16 years, while for those ≥65 it was 77.16 ± 6.31 years (*p* < 0.001). The group with the highest percentage of patients was 75–84 years, which included 36% of patients. Comorbidity was high (>3, assessed by the Charlson index), with no statistically significant differences between men and women (*p* = ns).

Regarding the degree of CKD severity, the highest percentage of patients presented a stage 4 chronic kidney disease (54.7%), 43.6% of patients presented DM, and 65.3% of the initial assessment was performed at the first outpatient visit at the ACKD unit. The general characteristics of the patients are shown in Table 1.

### 3.2. Results of the Functionality Study

The mean overall SPPB score was 8.50 ± 2.81. The mean value for this score was higher for men than for women (8.88 ± 2.49 vs. 7.77 ± 3.23, *p* = 0.005).

The cut-off point for good functional capacity in the SPPB test was established at 7 points. According to this cut-off, patients with a score ≥7 were considered to have a good functional capacity; these score values were observed in 76.4% of patients in our study.

According to the functional categories defined by the test, 76.5% of the patients had no limitations or mild limitations, while only 5.8% had severe limitations. Regarding the difference between men and women, the percentage of patients with severe or moderate limitations was significantly higher in women (*p* < 0.006). As the age ranges increased, the mean of the SPPB test result decreased, that is, the higher the age range, the lower the score on the test. The results of the SPPB test are shown in Table 2.

The mean SPPB score was higher in patients <65 years than in those ≥65 years (10.68 ± 1.60 vs. 7.58 ± 2.70, *p* < 0.001).

Figure 3 shows a comparison of the percentage of patients older and younger than 65 years within each functional category determined by assessment with the SPPB test.

As age increased, the percentage of patients with limitations increased; patients with an age <65 years did not present severe limitations, while only 1 patient with an age >85 years presented minimal/no limitations.

The results of the SPPB test according to age ranges are shown in Table 3.

The mean of the SPPB test was lower in frail patients, compared to non-frail patients (5.67 ± 2.49 vs. 9.11 ± 2.48, *p* < 0.001). This difference was similar in all age groups.

### 3.3. Association of Parameters of Nutritional Status and Body Composition with Patient Functionality

In those patients who presented a good functional capacity (SPPB ≥ 7), we found statistically significant better values for biochemical parameters such as the concentration of albumin, prealbumin, CRP, and creatinine, as well as for the nutritional status assessed by the MIS scale (Table 4).

In addition, we also found better values of body composition determined with bioimpedance and muscle strength assessed by dynamometry in those patients with SPPB ≥ 7 (Table 4). Regarding the rest of the parameters analyzed, we also observed better values in those patients who presented a good functional capacity, although the differences were not statistically significant.

In the multivariate analysis, logistic regression models showed that higher age, higher muscle strength, lean mass index, higher albumin (in model 1), and higher CRP (in model 2) were associated with an SPPB score ≥7 (Table 5).

## 4. Discussion

The objective of this study was to analyze the relationship between nutritional status and functional capacity (assessed by the SPPB test) in a group of patients with ACKD (non-dialysis). The results obtained showed that those patients with better nutritional status and body composition presented better scores in the SPPB test, thus confirming a relationship between nutritional status and functional capacity.

There are few studies assessing nutritional status and functional capacity in patients with ACKD not treated with renal replacement therapy (RRT) [26]; most studies have been carried out in patients in RRT, mainly in hemodialysis. Good nutritional status and functionality are crucial for patients with ACKD, since their survival at the beginning of RRT depends on these factors [27,28,29].

Progression of CKD and decrease in functional capacity have been previously linked [30,31,32]; however, in our study, we found no statistical difference in mean CKD-EPI between patients with good and bad functionality according to the SPPB cut-off point of 7. This may be because our study population was included in a protocol of assessment and monitoring to prevent functionality decrease.

Albumin was chosen as a nutritional marker and CRP as an inflammation marker (in addition to the rest of the parameters), since they are strongly associated with CKD progression, and they are both inversely interrelated when PEW is present [33,34]. These parameters should be analyzed together, since albumin is affected in inflammatory processes, while high CRP values are related to low albumin concentrations, and this association favors an increase in mortality of patients with CKD [35].

The study was conducted with a sample of patients with ACKD, 70.2% had an age ≥65 years (with a mean age within this group of 77.16 ± 6.31 years). Albumin concentration tends to decrease in elderly ACKD patients and when inflammation is present [36,37,38]. In our patients with ACKD s-albumin levels remain adequate despite they have advanced age. There are few patients with important inflammation, although these patients have decreased functionality. In our study, patients with s-albumin levels lower than 1 and s-CRP higher than 1 presented significantly lower SPPB scores.

The nutrition–inflammation status was assessed using MIS. This scale determines the nutritional status, establishing degrees of malnutrition according to score values. Patients with higher MIS scores have worse nutritional status. We have chosen the MIS Scale to diagnose PEW because, in addition to subjective parameters used in other scales, it also assesses objective parameters including albumin, thus providing added value to the diagnosis [39]. In this study, higher MIS values were associated with worse functional capacity.

In our study, we found that patients with MIS > 5 were included in the group with SPPB < 7 points, thereby indicating that higher malnutrition was associated with lower functionality.

We found, in relation to creatinine values, a good correlation with muscle mass parameters in the study of body composition by BIA and better functional capacity without significant difference in glomerular filtration rate measured by CKD-EPI. A small study with nine patients analyzing the effect of aerobic exercise found an increase in creatinine one month after performing a physical activity program, without a reduction in glomerular filtration [40].

Overall, age and CKD progression are high-risk factors for frailty [41] in these patients. Some authors found an association between reduced SPPB values and frailty and established a cut-off point of 8, for frailty so values <8 could be considered as a possible indicator of frailty [42,43]. Other studies have shown an association between a worse nutritional status and an increase in frailty [44,45,46].

Given the good correlation of SPPB with frailty as well as with age and CKD progression, some authors are studying the possibility of including SPPB as an indicator of frailty.

In 2021, Smith G et al. [47], studied the relationship between frailty assessed with the SPPB test, and markers strongly associated with CKD, such as creatinine and albumin, finding a relationship between frailty and functionality and both parameters. These results are in concordance with our findings that show an interrelationship between functionality, PEW, and frailty.

Therefore, from these results, we can affirm that diet will play a fundamental role, not only to control progression of renal disease through the control of protein intake [48], but also as a modulator of inflammation, which will influence the functionality of the patients. Accordingly, the Mediterranean diet, in addition to being a varied and balanced diet, may have, among other benefits, inflammation modulation properties [49]. These properties are related to its optimal monounsaturated and polyunsaturated fat profile, with an important contribution of omega-3 fatty acids [50], mainly EPA and DHA present in fish and vegetable oils and omega-6 fatty acids present in vegetable oils [50].

There is currently evidence of the usefulness of bioimpedance as a useful and simple tool to assess alterations of body composition [51] that can influence functionality (this technique provides information on fat-free mass and muscle mass, although through indirect measures). In our article, we observed significant differences between BIA parameters, which were better for patients with better functionality (SPPB > 7).

It is worth noting that this study was conducted in a group of patients from a multidisciplinary CKD unit with an established protocol for assessment, monitoring, and treatment, which facilitates the assessment of the patient nutritional state, body composition, muscle strength, and functional capacity on a regular basis and therefore facilitates body composition analysis (mainly muscle mass), as muscle strength can directly influence functional capacity [52,53,54]. These assessments are not readily available in all centers with ACKD units.

## 5. Conclusions

In conclusion, our results show that the SPPB test is a good tool for functionality assessment in ACKD patients and that nutritional status and body composition exert an influence on the functional capacity of these patients.

## Figures and Tables

**Figure 1 nutrients-14-04745-f001:**
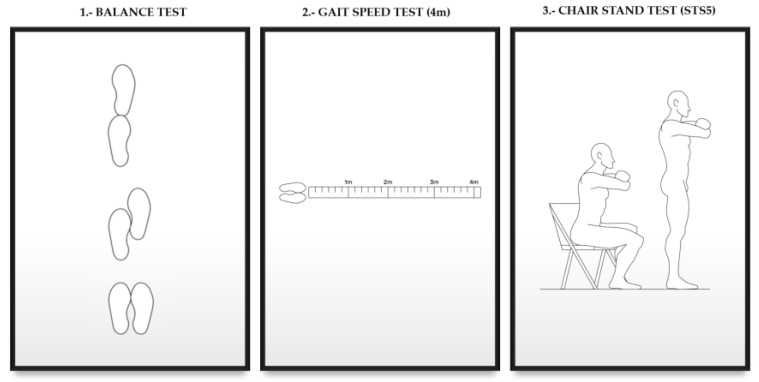
SPPB test.

**Figure 2 nutrients-14-04745-f002:**
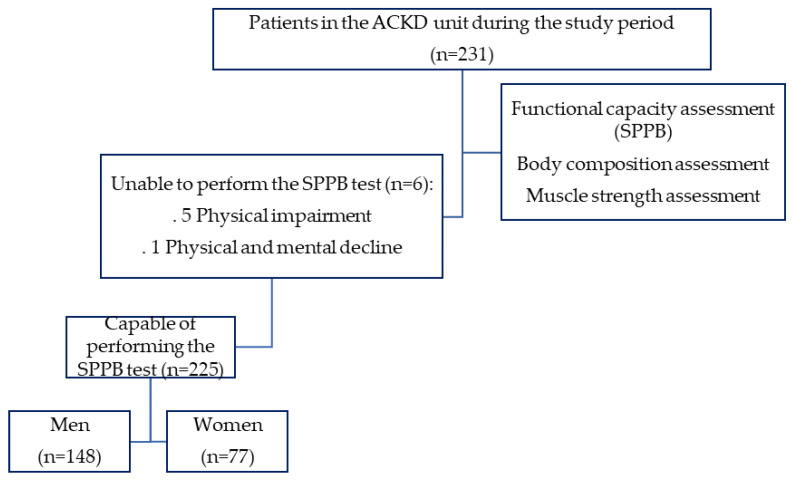
Study flowchart.

**Figure 3 nutrients-14-04745-f003:**
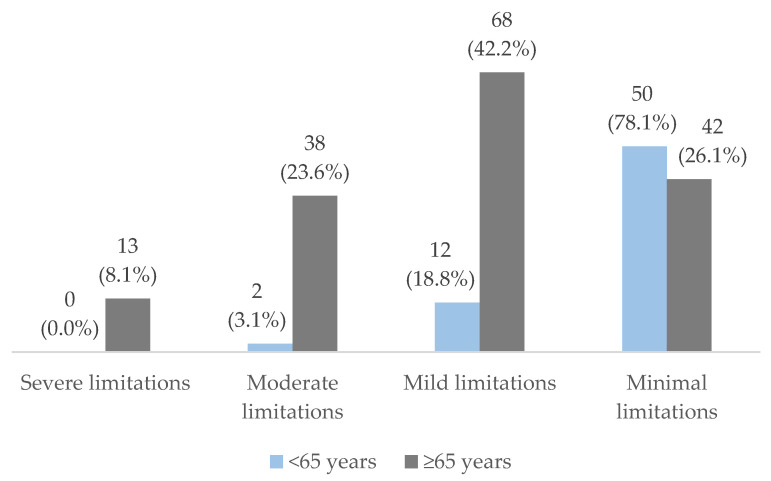
Comparison of percentage of patients <65 years and ≥65 years within each functional category assessed with the SPPB test (*p* < 0.001).

**Table 1 nutrients-14-04745-t001:** General characteristics of the study population.

	Overall	Men	Women	*p*-Value
Age years (mean ± SD/Median)	70.65 ± 11.97/73	69.85 ± 11.16/72	72.19 ± 13.33/76	ns
Sex *n* (%)	225	148 (65.8)	77 (34.2)	0.001
Age Range *n* (%)				0.012
<55	27 (12)	16 (10.8)	11 (14.3)	
55–64	36 (16)	27 (18.2)	9 (11.7)	
65–74	61 (27)	46 (31.1)	15 (19.5)	
75–84	81 (36)	52 (35.1)	29 (37.7)	
≥85	20 (9)	7 (4.7)	13 (16.9)	
ACKD stage *n* (%)				ns
Stage 3B	19 (8.4)	9 (6.1)	10 (13)	
Stage 4	123 (54.7)	87 (58.8)	36 (46.8)	
Stage 5 (ND)	83 (36.9)	52 (35.1)	31 (40.3)	
Comorbidity (mean ± SD/Median)	6.46 ± 1.92/6	6.56 ± 1.88/7	6.27 ± 1.99/6	ns
Frailty *n* (%)	40 (17.8)	20 (13.5)	20 (26%)	0.020
Time in ACKD unit *n* (%)				ns
<6 month	147 (65.3)	97 (65.5)	50 (64.9)	
6–12 month	29 (12.9)	21 (14.2)	8 (10.4)	
>12 months	49 (21.8)	30 (20.3)	19 (24.7)	
DM *n* (%)				0.016
Yes	98 (43.6)	73 (49.3)	25 (32.5)	
No	127 (56.4)	75 (50.7)	52 (67.5)	

Abbreviations: ACKD = advanced chronic kidney disease; ND = no dialysis; SD = standard deviation.

**Table 2 nutrients-14-04745-t002:** Short physical performance battery test.

	Global *n* = 225	Men 148	Women 77	*p*-Value
SPPB (mean ± SD/median)	8.50 ± 2.81/9	8.88 ± 2.49/9	7.77 ± 3.23/8	0.005
SPPB < 7 (low physical performance) *n* (%)	53 (23.6)	25 (16.9)	28 (36.4)	0.001
SPPB ≥ 7 (high physical performance) *n* (%)	172 (76.4)	123 (83.1)	49 (63.6)	
Severe limitations *n* (%)	13 (5.8)	4 (2.7)	9 (11.7)	0.006
Moderate limitations *n* (%)	40 (17.8)	21 (14.2)	19 (24.7)	
Mild limitations *n* (%)	80 (35.6)	57 (38.5)	23 (29.9)	
Minimal limitations *n* (%)	92 (40.9)	66 (44.6)	26 (33.8)	
Age Range (mean ± SD/median)				<0.001
<55	10.96 ± 1.31/11	10.87 ± 1.14/11	11.09 ± 1.57/12	
55–64	10.61 ± 1.79/11	10.88 ± 1.45/11	9.77 ± 2.48/11	
65–74	8.60 ± 2.34/9	8.47 ± 2.22/9	9.00 ± 2.72/9	
75–84	7.34 ± 2.62/7	7.88 ± 2.37/8	6.37 ± 2.82/6	
≥85	5.80 ± 2.83/5	6.71 ± 3.35/7	5.30 ± 2.52/5	

Abbreviations: SD, standard deviation; SPPB, short performance physical battery test. *p*-value of comparison men vs. women. Statistical significance *p* < 0.05.

**Table 3 nutrients-14-04745-t003:** Results of the SPPB according to age ranges.

	<55 Years	55–64 Years	65–74 Years	75–84 Years	≥85 Years	*p*-Value
Severe limitations *n* (%)	0 (0.0)	0 (0.0)	2 (3.3)	7 (8.6)	4 (20.0)	<0.001
Moderate limitations *n* (%)	0 (0.0)	2 (5.6)	8 (13.1)	22 (27.2)	8 (40.0)	
Mild limitations *n* (%)	5 (18.5)	6 (16.7)	28 (45.9)	34 (42.0)	7 (35.0)	
Minimal limitations *n* (%)	22 (81.5)	28 (77.8)	23 (37.7)	18 (22.2)	1 (5)	

Abbreviations: years: years.

**Table 4 nutrients-14-04745-t004:** Biochemical parameters according to SPPB cut-off point.

	Global *n* = 225 Mean ± SD	SPPB < 7 *n* = 53 Mean ± SD	SPPB ≥ 7 *n* = 172 Mean ± SD	*p*-Value
Albumin (g/dL)	4.20 ± 0.41	4.06 ± 0.39	4.26 ± 0.41	0.003
Prealbumin (mg/dL)	27.91 ± 7.74	25.91 ± 6.87	28.52 ± 7.93	0.041
CRP (mg/dL)	0.30 ± 1.40	1.13 ± 1.83	0.56 ± 1.22	0.010
Creatinine (mg/dL)	3.11 ± 1.31	2.98 ± 1.04	3.43 ± 1.37	0.026
Hemoglobin (g/dL)	12.10 ± 1.54	12.03 ± 1.43	12.31 ± 1.57	0.242
total lymphocytes (×10^3^/mm^3^)	1930 ± 931.38	1920.05± 946.96	2090.95 ± 925.60	0.244
Transferrin (mg/dL)	219.91 ± 51.60	211.28 ± 58.11	222.50 ± 49.25	0.171
GFR (CKD-EPI) (mL/min/1.73 m^2^)	17.73 ± 7.61	19.37 ± 8.68	18.61 ± 7.27	0.526
nPNA (g/kg weight/day)	0.86 ± 0.24	0.87 ± 0.20	0.92 ± 0.25	0.244
MIS	4.56 ± 2.99	2.89 ± 1.46	8.18 ± 2.14	<0.001
PA	4.23 ± 1.09	3.71 ± 1.07	4.39 ± 1.05	<0.001
Na/K	1.38 ± 0.44	1.50 ± 0.50	1.34 ± 0.41	0.021
%BCM	42.19 ± 8.18	38.07 ± 9.19	43.46 ± 7.43	<0.001
%IBW	43.44 ± 7.71	39.56 ± 8.53	44.64 ± 7.04	<0.001
%FM	31.22 ± 9.02	34.21 ± 10.39	30.30 ± 8.37	0.006
%FFM	68.77 ± 9.02	65.77 ± 10.40	69.69 ± 8.37	0.005
% MM	32.79 ± 7.71	29.83 ± 8.29	33.70 ± 7.32	0.001
BCMI	7.90 ± 2.01	6.72 ± 2.06	8.27 ± 1.85	<0.001
MMI	8.85 ± 2.02	7.85 ± 1.81	9.16 ± 1.99	<0.001
EMM	24.22 ± 7.27	20.25 ± 6.61	25.42 ± 7.04	<0.001
AMM	19.02 ± 4.74	16.28 ± 4.05	19.86 ± 4.62	<0.001
FFMI	18.62 ± 2.35	17.46 ± 2.12	18.97 ± 2.31	<0.001
%TBW	53.31 ± 7.39	51.85 ± 8.38	53.76 ± 7.03	0.099
FMI	8.89 ± 3.90	9.64 ± 4.38	8.66 ± 3.72	0.111
HGS	26.44 ± 10.60	19.07 ± 9.08	28.71 ± 10.00	<0.001

Abbreviations: SD = standard deviation; CRP = C-reactive protein; GFR = glomerular filtration rate; nPNA = normalized protein nitrogen appearance; MIS = malnutrition inflammation score; PA = phase angle; Na/K = Na/K ratio; BCM = body cell mass; IBW = intracellular body water; FM = fat mass; FFM = free fat mass; MM = muscle mass; BCMI = body cell mass index; MMI = muscle mass index; EMM = skeletal muscle mass; AMM = appendicular muscle mass; FFMI = fat-free mass index; TBW = total body water; FMI = fat mass index; HGS = handgrip strength.

**Table 5 nutrients-14-04745-t005:** Multivariate logistic regression.

	Model 1		Model 2	
	OR (95% CI)	*p*-Value	OR (95% CI)	*p*-Value
HGS (kg)	0.928 (0.879–0.981)	0.008	0.933 (0.888–0.979)	0.005
FFMI (kg/m^2^)	0.825 (0.882–0.997)	0.047	0.815 (0.675–0.984)	0.033
Albumin (g/dL)	0.245 (0.084–0.714)	0.010	-----	-----
Age (years)	1.111 (1.054–1.171)	<0.001	1.111 (1.059–1.166)	<0.001
CRP (mg/dL)	-----	-----	1.267 (1.007–1.594)	0.044

Abbreviations: OR = odds ratio; CI = confidence interval; HGS = handgrip strength; FFMI = fat-free mass index; CRP = C-reactive protein.

## Data Availability

Not applicable.

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
