# Peer review of "Impact of the Nutrition–Inflammation Status on the Functionality of Patients with Chronic Kidney Disease"

_nutrients, 2022, doi:10.3390/nu14224745_

Round 1
Reviewer 1 Report
This manuscript mainly studied the functional ability of patients with chronic kidney disease and the influence of nutritional inflammation on kidney functional ability. The author selected SPPB test to evaluate the functional capacity of patients with advanced chronic kidney disease. This manuscript has some significance for the study of kidney diseases. There are some comments for the authors.
1. Figure 2, the line number is misaligned and the figure needs to be modified.
2. Line 432, the reference is incorrectly formatted.
3. The manuscript format is not standardized. For example, some paragraphs are justified, and some paragraphs are left justified; the format is not uniform in keywords; the format of line 39 references is different from that of other references.
4. What do A and B represent in the abstract, please indicate.
Author Response
Revisor 1
This manuscript mainly studied the functional ability of patients with chronic kidney disease and the influence of nutritional inflammation on kidney functional ability. The author selected SPPB test to evaluate the functional capacity of patients with advanced chronic kidney disease. This manuscript has some significance for the study of kidney diseases. There are some comments for the authors.
Thank you very much for your suggestions.
- Figure 2, the line number is misaligned and the figure needs to be modified:
The figuere has been modified.
- Line 432, the reference is incorrectly formatted.
The reference has been changed in the manuscript
- The manuscript format is not standardized. For example, some paragraphs are justified, and some paragraphs are left justified; the format is not uniform in keywords; the format of line 39 references is different from that of other references.
The manuscript format has been standarized in the new manuscript
- What do A and B represent in the abstract, please indicate.
We do not know what is A and B in the manuscript, please could you be more specific, about this point? We send two mail in relation with this and don´t received answer. We can correct this ítem in 24h when we received instruction.
Reviewer 2 Report
Dear authors
Your article entitled "Impact of the nutrition-inflammation status on the functionality of patients with Chronic Kidney Disease" is extremely interesting. The conducted research proves how much nutrition and nutritional status contribute to the course and development of chronic diseases. Dietary restrictions in chronic kidney disease foster nutrient deficiencies in the body. The presented work is of great importance both for dietitians and the medical service. Both of the above teams of specialists are two complementary tools to help patients. I believe that the manuscript has enormous practical potential, and the research results should be made available as soon as possible.
Author Response
Your article entitled "Impact of the nutrition-inflammation status on the functionality of patients with Chronic Kidney Disease" is extremely interesting. The conducted research proves how much nutrition and nutritional status contribute to the course and development of chronic diseases. Dietary restrictions in chronic kidney disease foster nutrient deficiencies in the body. The presented work is of great importance both for dietitians and the medical service. Both of the above teams of specialists are two complementary tools to help patients. I believe that the manuscript has enormous practical potential, and the research results should be made available as soon as possible.
Thank you very much for your comments, we are very happy to know that this work it is interesting for you and we will be glad to submited it, as son as posible.
Round 2
Reviewer 1 Report
This reviewer accepts the revised version